# Funnel-Transformer: Filtering out Sequential Redundancy for Efficient Language Processing

**Zihang Dai**[*12], **Guokun Lai**[*1], **Yiming Yang**[1], **Quoc V. Le**[2]
[1]Carnegie Mellon University, [2]Google AI Brain Team
{dzihang,guokun,yiming}@cs.cmu.edu, qvl@google.com

## Abstract

With the success of language pretraining, it is highly desirable to develop more efficient architectures of good scalability that can exploit the abundant unlabeled data at a lower cost. To improve the efficiency, we examine the much-overlooked redundancy in maintaining a full-length token-level presentation, especially for tasks that only require a single-vector presentation of the sequence. With this intuition, we propose Funnel-Transformer which gradually compresses the sequence of hidden states to a shorter one and hence reduces the computation cost. More importantly, by re-investing the saved FLOPs from length reduction in constructing a deeper or wider model, we further improve the model capacity. In addition, to perform token-level predictions as required by common pretraining objectives, Funnel-Transformer is able to recover a deep representation for each token from the reduced hidden sequence via a decoder. Empirically, with comparable or fewer FLOPs, Funnel-Transformer outperforms the standard Transformer on a wide variety of sequence-level prediction tasks, including text classification, language understanding, and reading comprehension.[1]

## 1 Introduction

With the recent success of unsupervised language pretraining [1, 2, 3, 4, 5, 6, 7, 8, 9, 10, 11, 12], the power of neural self-attention models (a.k.a. Transformer) [13] has been pushed to a new level, leading to dramatic advancements in machine learning and natural language processing (NLP). More importantly, it has been observed that with more FLOPs invested in longer pretraining and/or larger models, the performance of pretrained Transformer models consistently improve. However, it is extremely expensive to pretrain or even just to finetune the state-of-the-art self-attention models, as they require much more FLOPs and memory resources compared to traditional models in NLP. This largely limits their applications and success in more fields.

Given this challenge, there has been an increasing amount of efforts to reduce the costs of pretraining and finetuning self-attention models. From the perspective of post-pretraining processing, typical approaches include distillation, pruning and quantization of various kinds, which try to derive a lighter model from an well-pretrained model by taking advantage of the richer signals in the larger model or learning to remove less important operations. Another line of research aims at designing an architecture that not only has a lower resource-to-performance ratio (more efficient) but also *scales as well as* the Transformer, at least in certain domains. Most of such methods build upon the Transformer backbone and focus on redesigning its building blocks. Representative solutions include searching for better micro operation or macro module designs [14, 15], replacing the full pairwise attention with local operations such as convolution [16] and dynamic convolution [17], and optimizing the hidden size combinations for existing blocks [18].

---

[*]Equal contribution.
[1]The code and pretrained checkpoints are available at github.com/laiguokun/Funnel-Transformer.

Across the wide variety of ideas mentioned above, a common strategy is to identify redundant operations or representations and replace them with more efficient ones. Inspired by this line of thinking, in this work, we will be focusing on the potential redundancy induced by always maintaining a *full-length sequence* of hidden representations across all layers in Transformer. Intuitively, for many sequence-level NLP tasks such as text classification and ranking, the most common use case is to extract a *single* vector from the entire sequence, which does not necessarily preserve all information down to the token-level granularity. Hence, for such tasks, the full-length sequence of hidden states may contain significant redundancy. This is analogous to the case of image recognition, where the convolution neural network gradually reduces the spatial resolution/size of feature maps as the neural network goes deeper. In addition, linguistic prior also encourages gradually merging nearby tokens (words) into larger semantic units (phrases), which naturally leads to a shorter sequence of representations.

Concretely, we propose to gradually reduce the sequential resolution (i.e. length) of the hidden representation in self-attention models. Immediately, the reduction in sequence length can lead to significant savings in both FLOPs and memory. More importantly, the saved computational resource can be directly re-invested in constructing a deeper (or wider) model to boost the model capacity without additional computational burden. In addition, to address the challenge that common pretraining objectives such as masked language modeling (MLM) [2] require separate representations for each token, we design a simple strategy to decode a full-length sequence of deep representations from the hidden state of reduced length. As a result, the proposed model can be directly trained without modifying the pretraining objectives, as well as adopted for downstream tasks that require token-level representations.

Empirically, with comparable or even fewer FLOPs, by trading sequential resolution for depth, our proposed model achieves an improved performance over the standard Transformer on a wide variety of sequence-level prediction tasks, including text classification, language understanding, and reading comprehension.

## 2  Method

### 2.1  Background

**Transformer Architecture**   The Transformer architecture [13] is a highly modularized neural network, where each Transformer layer consists of two sub-modules, namely the multi-head self-attention (S-Attn) and position-wise feed-forward network (P-FFN). Both sub-modules are wrapped by a residual connection and layer normalization. Schematically, given a length $T$ sequence of hidden states $\mathbf{h} = [h_1, \ldots, h_T]$, the computation of a single Transformer layer can be expressed as

$$\mathbf{h} \leftarrow \text{LayerNorm}(\mathbf{h} + \text{S-Attn}(\mathbf{Q} = \mathbf{h}, \mathbf{KV} = \mathbf{h})), \tag{1}$$

$$h_i \leftarrow \text{LayerNorm}(h_i + \text{P-FFN}(h_i)), \quad \forall i = 1, \cdots, T. \tag{2}$$

**Pretraining Objectives**   The most commonly used pretraining objective is the masked language modeling (MLM) proposed by BERT [2]. For a length-$T$ natural language sequence $\mathbf{x}$ sample from a large unlabeled set $\mathcal{D}$, the MLM objective first constructs a corrupted sequence $\hat{\mathbf{x}}$ by randomly replacing 15% of the tokens of $\mathbf{x}$ with a special token [mask] and then trains a Transformer model [2] to reconstruct the original $\mathbf{x}$ based on $\hat{\mathbf{x}}$, i.e.,

$$\max_{\theta} \ \mathcal{J}_{\text{MLM}}(\theta) = \mathbb{E}_{\mathbf{x} \sim \mathcal{D}} \mathbb{E}_{\mathcal{I}} \sum_{i \in \mathcal{I}} \log P_\theta(x_i \mid \hat{\mathbf{x}}_{\mathcal{I}}) = \mathbb{E}_{\mathbf{x} \sim \mathcal{D}} \mathbb{E}_{\mathcal{I}} \sum_{i \in \mathcal{I}} \log \frac{\exp \left( e(x_i)^\top h_i(\hat{\mathbf{x}}_{\mathcal{I}}) \right)}{\sum_{x'} \exp \left( e(x')^\top h_i(\hat{\mathbf{x}}_{\mathcal{I}}) \right)},$$

where $\mathcal{I}$ is the positions of masked tokens, the subscript in $\hat{\mathbf{x}}_{\mathcal{I}}$ emphasizes its dependence on $\mathcal{I}$, $e(x)$ denotes the embedding of the token $x$, and $h_i(\hat{\mathbf{x}}_{\mathcal{I}})$ the last-layer hidden state at position $i$ produced by the Transformer model. After pretraining, the entire model is finetuned in downstream tasks.

To show the generality of our proposed model, we also experiment with another pretraining objective ELECTRA [5]. Different from MLM, ELECTRA relies a pair of jointly trained generator and discriminator. Specifically, the generator usually has a smaller size (1/4 of that of the discriminator) and is directly trained via the MLM objective, i.e., $\max_{\theta_G} \mathcal{J}_{\text{MLM}}(\theta_G)$. Then, for each masked position, a token is sampled from the reconstruction distribution of the generator to replace the [mask] token and form a new sequence $\tilde{\mathbf{x}}$, i.e., if $i \in \mathcal{I}, \tilde{x}_i \sim P_{\theta_G}(x_i \mid \hat{\mathbf{x}}_{\mathcal{I}})$ else $\tilde{x}_i = x_i$. Given the new sequence $\tilde{\mathbf{x}}$, the discriminator is then trained to distinguish whether each token in $\tilde{\mathbf{x}}$ is real (same as $\mathbf{x}$) or fake (different from $\mathbf{x}$) via binary classification. After pretraining, only the discriminator will be used during finetuning and the generator is simply discarded.

**Discussion** Note that both pretraining objectives introduced above require the ability to produce a hidden state for each input token, i.e., $h_i(\hat{\mathbf{x}}_{\mathcal{I}})$ and $h_i(\tilde{\mathbf{x}})$. Due to this requirement, it seems natural to keep a full sequence of hidden states. However, in contrast, many sequence-level downstream tasks like classification or ranking only need a single-vector summary of the entire sequence. Fundamentally, this suggests that some kind of compression is usually required to remove the unnecessary redundancy during finetuning. This observation immediately leads to the following two questions:

- Can we design a general model that is equally expressive but more efficient by compressing the full sequence of hidden states into a more compact form?
- With the compressed representations, how can the model retain the ability to produce token-level representations for pretraining?

To answer these two questions, we next present our proposed architecture.

## 2.2 Proposed Architecture

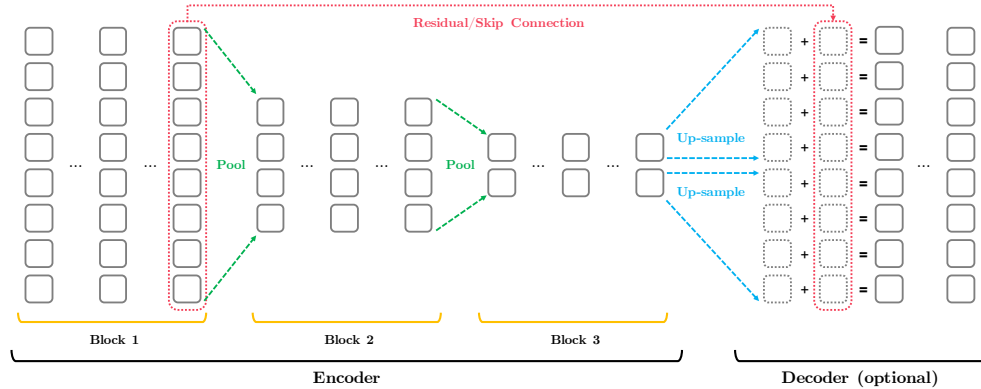

Figure 1: High-level visualization of the proposed Funnel-Transformer.

To inherit the high capacity and optimization advantages of the Transformer architecture, the proposed model keeps the same overall skeleton of interleaved S-Attn and P-FFN sub-modules wrapped by residual connection and layer normalization. But differently, to achieve representation compression and computation reduction, our model employs an encoder that gradually reduces the sequence length of the hidden states as the layer gets deeper. In addition, for tasks involving per-token predictions like pretraining, a simple decoder is used to reconstruct a full sequence of token-level representations from the compressed encoder output.

**Encoder** As illustrated in the left part of Fig. 1, the encoder consists of several blocks of consecutive Transformer layers. Within each block, the sequence length of the hidden states always remains the same. But when going from a lower-level block to a higher-level block, the length of the hidden sequence is reduced by performing certain type of pooling along the sequence dimension, i.e.,

$$\mathbf{h}' \leftarrow \text{Pooling}(\mathbf{h}), \tag{3}$$

where $\mathbf{h} \in \mathbb{R}^{T \times D}$ and $\mathbf{h}' \in \mathbb{R}^{T' \times D}$ for some $T' < T$. Importantly, instead of directly feeding the pooled sequence $\mathbf{h}'$ into the first S-Attn layer of the new block, we only use pooled sequence $\mathbf{h}'$ to construct the query vector (and the residual signal) of the self-attention, while the unpooled sequence $\mathbf{h}$ serves that role of key and value vectors, i.e.

$$\mathbf{h} \leftarrow \text{LayerNorm}\big(\mathbf{h}' + \text{S-Attn}(\mathbf{Q} = \mathbf{h}', \mathbf{KV} = \mathbf{h})\big). \tag{4}$$

Note that the output sequence of this special S-Attn module has the same length as the pooled sequence $\mathbf{h}'$. To understand the advantage of this particular design, it is helpful to compare the proposed "pool-query-only" variant with the naive alternative of using $\mathbf{h}'$ for both the query and key-value vectors, i.e., S-Attn$\big(\mathbf{Q} = \mathbf{h}', \mathbf{KV} = \mathbf{h}'\big)$:

- Under the naive approach, the compression is solely controlled by the pooling operation, which is finished before the attention module. Hence, relatively simple pooling methods such as average/mean pooling won't be able to achieve good compression.
- Under the pool-query-only variant, the compression depends on not only how the pooling is performed, but also how the self-attention weighted sums the unpooled sequence to form each

pooled vector. Effectively, the particular attention here can be seen as a type of linear compression that combines $T$ bases into a smaller number of $T'$ "compressed bases". Therefore, with minimum computational overhead, this variant makes compression operation more expressive.

With this particular pool-query-only design in place, we find the simplest strided mean pooling applied to each sliding window of the sequence work very well in practice. For simplicity, we only experiment with stride 2 and window size 2 in this work. Hence, the pooling operation will reduce the sequence by half and each pooled hidden state corresponds to a window of 2 unpooled hidden vectors. Intuitively, this type of pooling roughly follows the linguistic prior that nearby tokens could be gradually merged (or compressed) into a larger semantic component. Once the sequence length is halved after the pooling and pool-query-only attention, the rest of the encoder computation simply follows the standard updates in Eqn. (2) and (1).

Finally, as an extra implementation detail, recall that a particular design in language pretraining is to add a special token [cls] to the beginning of the original input sequence, and use the last-layer hidden state corresponding to [cls] (i.e., $h_1$) as the representation of the sequence. To prevent the pooling from destroying this special structure, we first separate the [cls] hidden state and the rest of hidden states and only apply the pooling to the rest of hidden states. For some practical implementation issues and an efficient solution, we refer readers to Appendix A.1.

**Decoder** In order to recover a full sequence of hidden states from the encoder output of reduced length, a natural idea would be performing some kind of up-sampling. For instance, in image generation or super-resolution, deconvolution (transposed convolution) or parameter-free resizing with bilinear interpolation are often used to increase the spatial resolution of the feature map. Hence, we can simply adapt these ideas from 2D processing to our 1D case and apply proper up-sampling to the encoder output.

However, instead of performing multiple up-samplings with small expansion rate (e.g. increasing the sequence length by 2x each time) as in image domain, we here choose to employ a single up-sampling with a large expansion rate, as shown on the right part of Fig. 1. Specifically, given the output sequence $\mathbf{h^M}$ of length $T_M = T/2^{M-1}$ from an $M$-block encoder, we directly up-sample it to a full-length sequence $\mathbf{h^{up}} = \left[h_1^{up}, \cdots, h_T^{up}\right]$ by repeating each hidden vector $2^{M-1}$ times:

$$\forall i = 1, \cdots, T, \quad h_i^{up} = h_{i//2^{M-1}}^M, \tag{5}$$

where $\cdot // \cdot$ denotes floor division. However, note that every $2^{M-1}$ consecutive vectors in $\mathbf{h^{up}}$ are exactly the same and hence do not contain detailed token-level information. Hence, we further extract the last-layer hidden states from the *first block* of the encoder $\mathbf{h^1}$, which still has the full length $T$ and contains the uncompressed token-level information. Then, the lower-level representation $\mathbf{h^1}$ and up-sampled higher-level representation $\mathbf{h^{up}}$ are added together to form a deep token-level representation $\mathbf{g} = \mathbf{h^1} + \mathbf{h^{up}}$. Effectively, this forms a residual/skip connection that enables detailed token information and potentially easier optimization. In addition, we stack a few more Transformer layers upon $\mathbf{g}$ to achieve a better deep fusion of the low-level and high-level features. In this work, we always use 2 Transformer layers in decoder.

It is important to emphasize that the decoder is *only used if the task requires token-level prediction*, such as in standard pretraining or sequence labeling. For tasks that only requires a single vectorial representation of the sequence like classification, the decoder is discarded after pretraining and only the encoder is finetuned. Finally, to emphasize the filtering/compression property of the encoder as well as its shape, we name the proposed model `Funnel-Transformer` (F-TFM).

### 2.3 Complexity & Capacity Analysis

With the architecture design specified, we now analyze how the sequence compression affects the complexity and capacity of the proposed model, especially compared to the standard Transformer.

Firstly, for a Transformer layer with an S-Attn and a P-FFN module of hidden size $D$, the complexity of processing a length-$T$ sequence is $O(T^2D + TD^2)$.[2] Hence, every time the sequence length is reduced by half in the encoder, we enjoy a *super-linear* (more than half) complexity drop. In practice, as the $O(TD^2)$ term has a large constant, a near-linear speedup is observed more often. The super-linear effect is more detectable when the sequence length is relatively long like in pretraining.

Therefore, given the same FLOPs, we can at least trade a full-length layer in the 1st block for $2^{m-1}$ layers in the $m$-th block, which provides an economical way to increase the depth of network.

On the other hand, the capacity of a compressed-length layer is clearly upper-bounded by that of a normal full-length layer. In most cases where the compression is lossy, reducing the sequence length will inevitably lead to capacity drop. The good news is that the capacity drop of a single layer could be well compensated by re-investing the saved FLOPs in stacking more cheaper layers of reduced length or increasing the width of the model.

As a concrete example, for a Transformer of BERT$_{\text{Base}}$ size, i.e., 12 layers of hidden size 768 (L12H768), we may construct a Funnel-Transformer of 3 blocks where each block has 6 layers of hidden size 768 (B6-6-6H768). Despite having 18 layers in total, when finetuned for classification, the FLOPs of the B6-6-6H768 architecture only corresponds to at most $6 + 6/2 + 6/4 = 10.5$ full-length layers, clearly fewer than that of L12H768. More importantly, as we will show in the experiments, B6-6-6H768 significantly outperforms L12H768. While intuitive, how to construct an optimal block layout given this *depth-length trade-off* remains an open challenge. For this work, we only consider relatively regular layout and leave more systematic studies for future work.

Finally, notice that trading sequential resolution for depth or width has a side effect of increasing the total number of parameters. For instance, B6-6-6H768 has 1.5x Transformer parameters compared to L12H768. In practice, more parameters may increase communication cost in distributed training as well as the memory consumption and memory access time. A simple remedy is to perform certain parameter sharing, as used in ALBERT, to recover the same parameter count. Taking B6-6-6H768 as an example, one may tie the parameters for every two layers in the 2nd and 3rd blocks, denoted as B6-3x2-3x2H768, which gives back the same number of parameters to L12H768. However, parameter sharing could result in performance loss. Fundamentally, this brings us another trade-off between the gain (capacity) and cost (memory and communication cost) of using more parameters, which can be highly device dependent.

## 3   Related Work

As far as we know, no previous work achieves performance gain via compressing the sequence length of the hidden states under language pretraining. Meanwhile, our proposed model is quite similar to the bottom-up model proposed by a contemporary work [19] for causal language modeling. The key differences include the pool-query-only design for down-sampling, how the up-sampling is performed, and our relative attention parameterization. Another closely related idea is Power-BERT [20], which learns to *soft-eliminate* word vectors that are less "significant" during finetuning. Hence, for post-finetuning inference, the sequence length can be reduced to achieve acceleration. More generally, our work is also related to previous work on hierarchical recurrent neural networks [21] and Transformer models [22, 23]. Different from these methods, our model does not rely on any pre-defined hierarchy or boundary of semantic meanings and always captures the full-length dependency input with attention.

In contrast, our work draws many inspirations from the computer vision domain. The contracting encoder and expanding decoder framework with residual connections is conceptually similar to the ResUNet [24] for image segmentation. The strided pooling is also widely used to construct modern image recognition networks [25]. Despite the similarities, apart from the obvious difference in data domain and computation modules, our encoder employs a special pool-query-only design to improve the compression, and our decoder only requires a single up-sampling with a large expansion rate.

In addition, a line of research in graph neural networks has tries to gradually reduce the number of nodes in different ways and obtain a single vectorial representation for supervised classification. [26, 27, 28] While these methods could potentially be plugged into our model as alternative compression operations, it remains an open question whether compression techniques developed for supervised graph classification can be extended the large-scale language pretraining.

## 4   Experiment

In this section, we empirically evaluate the proposed F-TFM by first pretraining it and then finetuning it in downstream tasks. Following previous work, for pretraining, we consider two common settings:

- **Base scale**: Pretraining models for 1M steps with batch size 256 on Wikipedia + Book Corpus. This is the setting used by original BERT [2]. We will rely on this setting to perform fair comparison between F-TFM and the standard Transformer as well as some ablation studies.

- **Large scale**: Pretraining models for 500K steps with batch size 8K on the five datasets used by XLNet [3] and ELECTRA [5] (Wikipedia + Book Corpus + ClueWeb + Gigaword + Common Crawl). We will compare F-TFM trained at this scale with previous state-of-the-art methods.

For finetuning, we mainly focus on sequence-level tasks that only requires a single vectorial representation of the input sequence, since F-TFM is designed with such a purpose in mind. Specifically, such tasks include the GLUE benchmark for language understanding [29, 30, 31, 32, 33, 34, 35, 36, 37], 7 widely used text (sentiment / topic) classification tasks (IMDB, AD, DBpedia, Yelp-2, Yelp-5, Amazon-2, Amazon-5) [38], and the RACE reading comprehension dataset [39]. In addition, to see how F-TFM performs when token-level prediction is needed, we consider the SQuAD question answering task which requires the model to select a token span from the context paragraph as the answer. For more details of the experiment setting, we refer readers to Appendix B.

Finally, for all models implemented in this work including Transformer baselines in the base-scale comparison section 4.1, we always use the relative positional attention parameterization proposed by Transformer-XL [40] (see Appendix A.2 for some implementation details of Transformer-XL).

### 4.1  Base-scale Results

Firstly, we evaluate how F-TFM performs compared to the standard Transformer under similar amount of computation (i.e., FLOPs). For this purpose, we consider three commonly used model sizes for the standard Transformer, namely large (L24H1024), base (L12H768) and small (L6H768). Then, for each Transformer baseline, we construct F-TFMs of different block layouts and parameters, while ensuring the F-TFMs always have fewer or similar FLOPs. Based on the MLM pretraining objective, the results on GLUE benchmark and text classification are presented in Table 1, where we also include the *relative* FLOPs and #Params. Here, we can make a few key observations:

| Model size | CoLA | SST-2 | MRPC | STS-B | QQP | MNLI | QNLI | RTE | GLUE-AVG |
|---|---|---|---|---|---|---|---|---|---|
| L24H1024 | 63.2 | 94.8 | 91.8/88.5 | 91.1 | 88.7/91.7 | 88.7 | 94.0 | 80.5 | 86.6 |
| B10-10-10 | 64.8 | 95.0 | 92.5/89.5 | 90.7 | 88.6/91.5 | 88.9 | 94.0 | 81.5 | **87.0** |
| B8-8-8 | 63.5 | 94.7 | 92.2/89.0 | 90.7 | 88.9/91.7 | 88.8 | 93.6 | 81.2 | 86.7 |
| L12H768 | 60.5 | 93.0 | 92.2/89.0 | 89.4 | 88.1/91.2 | 86.0 | 92.2 | 73.6 | 84.4 |
| B6-6-6 | 62.5 | 94.0 | 92.2/89.0 | 89.5 | 88.4/91.4 | 87.0 | 92.7 | 76.5 | **85.3** |
| B6-3x2-3x2 | 60.5 | 93.6 | 92.4/89.2 | 89.4 | 88.2/91.3 | 86.4 | 92.5 | 75.0 | 84.7 |
| B4-4-4 | 59.1 | 92.7 | 91.8/88.7 | 89.1 | 88.2/91.3 | 85.5 | 92.0 | 73.2 | 83.9 |
| L6H768 | 55.2 | 91.5 | 91.1/87.8 | 88.1 | 87.2/90.6 | 82.7 | 90.0 | 64.6 | 81.3 |
| B3-4-4 | 59.0 | 92.8 | 91.8/88.5 | 88.5 | 87.8/90.9 | 84.8 | 91.8 | 73.2 | **83.7** |

| Model size | IMDB | AG | DBpedia | Yelp2 | Yelp5 | Amazon2 | Amazon5 | FLOPs | #Params |
|---|---|---|---|---|---|---|---|---|---|
| L24H1024 | 4.440 | **4.987** | 0.646 | 1.758 | 28.73 | 2.409 | 32.78 | 1.00x | 1.00x |
| B10-10-10 | **4.404** | 5.026 | **0.617** | **1.734** | **28.52** | **2.400** | **32.65** | 0.73x | 1.22x |
| B8-8-8 | 4.552 | 5.079 | 0.664 | 1.713 | 28.84 | 2.438 | 32.87 | 0.58x | 1.00x |
| L12H768 | 5.328 | 5.184 | 0.663 | 2.013 | 29.35 | 2.571 | 33.14 | 1.00x | 1.00x |
| B6-6-6 | **4.908** | **5.079** | 0.654 | **1.939** | **29.03** | **2.518** | **32.91** | 0.88x | 1.39x |
| B6-3x2-3x2 | 5.144 | 5.342 | **0.649** | 1.892 | **29.03** | 2.570 | 33.01 | 0.88x | 1.00x |
| B4-4-4 | 5.348 | 5.250 | 0.670 | 1.979 | 29.37 | 2.596 | 33.16 | 0.58x | 1.00x |
| L6H768 | 6.252 | 5.421 | 0.697 | 2.203 | 30.33 | 2.801 | 33.69 | 1.00x | 1.00x |
| B3-4-4 | **5.520** | **5.342** | **0.670** | **2.042** | **29.51** | **2.603** | **33.16** | 1.00x | 1.53x |

Table 1: MLM pretraining results at the base scale: GLUE dev *performances (the higher the better)* in the upper panel and text classification *error rates (the lower the better)* in the lower panel . The FLOPs and #Params both refer to the finetuning setting with only the encoder. The corresponding numbers with the decoder are included in Appendix C.2. The FLOPs is a rough estimation assuming linear complexity w.r.t. the sequence length. The #Params is exact including the embedding matrix.

- Given similar or fewer FLOPs, by trading sequential resolution for more layers, the F-TFM outperforms the standard Transformer in most tasks except STS-B, especially for smaller models.
- When we only compress the sequence length without increasing the depth (and #Params), F-TFM could suffer from some performance loss in certain settings on the GLUE datasets. However, as the model size increases, such performance gaps become smaller or even disappear.
- In addition, we find partial parameter-sharing often harms the performance. Therefore, the practical trade-off should be made according to the actual task and computation device.

To further test generality of F-TFM, we additionally consider ELECTRA for pretraining. The results are summarized in Table 2. Overall, we see a similar trend, though the gain is slightly smaller on the GLUE benchmark. This could be attributed to reusing two key hyper-parameters (discriminator loss coefficient and generator size multiplier) tuned for Transformer to train F-TFMs without any adjustment at all.

| Model size | CoLA | SST-2 | MRPC | STS-B | QQP | MNLI | QNLI | RTE | GLUE-AVG |
|---|---|---|---|---|---|---|---|---|---|
| L24H1024 | 66.5 | 94.3 | 92.8/90.0 | 91.5 | 89.6/92.2 | 89.4 | 94.1 | 84.5 | 87.8 |
| B10-10-10 | 68.6 | 95.0 | 93.0/90.0 | 91.0 | 88.9/91.7 | 89.1 | 93.6 | 84.5 | **87.9** |
| B8-8-8 | 66.6 | 94.8 | 92.6/89.7 | 90.7 | 88.8/91.7 | 89.0 | 93.6 | 82.1 | 87.3 |
| L12H768 | 64.3 | 93.1 | 92.1/89.2 | 90.8 | 88.7/91.7 | 86.4 | 92.1 | 75.4 | 85.4 |
| B6-6-6 | 64.3 | 94.2 | 92.8/89.7 | 90.1 | 88.7/91.6 | 87.4 | 92.5 | 78.3 | **86.0** |
| B6-3x2-3x2 | 63.9 | 94.2 | 93.0/90.2 | 89.5 | 88.4/91.4 | 87.0 | 92.2 | 77.6 | 85.7 |
| B4-4-4 | 62.8 | 93.6 | 92.5/89.2 | 89.2 | 88.4/91.3 | 86.0 | 91.6 | 74.3 | 84.8 |
| L6H768 | 62.1 | 91.1 | 90.8/86.8 | 88.9 | 88.2/91.3 | 83.9 | 89.7 | 66.7 | 82.6 |
| B3-4-4 | 59.0 | 93.1 | 90.8/87.5 | 88.7 | 88.1/91.0 | 85.8 | 91.1 | 72.5 | **83.6** |

| Model size | IMDB | AG | DBpedia | Yelp2 | Yelp5 | Amazon2 | Amazon5 | FLOPs | #Params |
|---|---|---|---|---|---|---|---|---|---|
| L24H1024 | 4.724 | **5.053** | 0.653 | 1.874 | 28.84 | 2.425 | 32.85 | 1.00x | 1.00x |
| B10-10-10 | **4.324** | 5.250 | **0.639** | **1.789** | **28.68** | **2.419** | **32.72** | 0.73x | 1.22x |
| B8-8-8 | 4.364 | 5.408 | 0.651 | 1.729 | 28.76 | 2.447 | 32.85 | 0.58x | 1.00x |
| L12H768 | 5.248 | 5.355 | 0.657 | 1.953 | 29.24 | 2.596 | 33.04 | 1.00x | 1.00x |
| B6-6-6 | **4.792** | **5.237** | **0.650** | **1.850** | **28.73** | **2.499** | **32.79** | 0.88x | 1.39x |
| B6-3x2-3x2 | 4.924 | 5.342 | 0.671 | 1.913 | 29.00 | 2.523 | 32.85 | 0.88x | 1.00x |
| B4-4-4 | 5.152 | 5.382 | 0.659 | 2.032 | 29.33 | 2.566 | 33.03 | 0.58x | 1.00x |
| L6H768 | 6.220 | 5.395 | 0.674 | 2.287 | 30.16 | 2.759 | 33.57 | 1.00x | 1.00x |
| B3-4-4 | **5.396** | **5.342** | **0.653** | **2.000** | **29.60** | **2.591** | **33.09** | 1.00x | 1.53x |

Table 2: ELECTRA pretraining results at the base scale.

**Running Time Comparison** While FLOPs count offers a general idea of the model speed, it still differs from the actual running time, especially when other overhead exists. Hence, for completeness, we show the speedup provided by the F-TFM in terms of actual running time in Appendix C.2. We also compare the actual memory footprint of F-TFM and TFM in Appendix C.2.

## 4.2 Large-scale Results

Given the encouraging results of F-TFM at base-scale, we next consider training F-TFM under the large-scale setting and compare it with previous models pretrained in similar settings. Due to the slightly better performance of ELECTRA over MLM, we will use the ELECTRA objective for all large-scale experiments. Given the pretrained F-TFM of different sizes, we first compare the finetuning performance on the GLUE benchmark in Table 3. Similar to the base-scale results, with fewer or comparable FLOPs, F-TFM outperforms the corresponding baselines in the majority of tasks, suggesting the good scalability of F-TFM. We also test the models on the 7 text classification tasks. But due to the page constraint, we refer readers to Appendix C.1.

Next, we consider the RACE dataset, which is quite different from the GLUE benchmark. At the core, RACE is a multiple-choice reading comprehension task requiring complex reasoning, which though, can be formulated as classifying the correct choice. Also, paragraphs in RACE are much longer. To F-TFM, this presents both a challenge, as it requires detailed reasoning, and an opportunity to compress long paragraph. As we can see in Table 4, F-TFM achieves better performances compared to all previous models. In particular, within the base model group, the gain is very significant. It shows that F-TFM can also excel for sequence-level task that involves long text and reasoning.

Finally, although F-TFM is mainly designed for tasks that only require a sequence-level representation, it is possible to apply F-TFM to token-level tasks by additionally finetuning the decoder. To test this ability, we finetune F-TFM on the SQuAD datasets and compare it with previous models in Table 5. While F-TFM outperforms previous models in the base group by a large margin, in the large model group, the F-TFM with about 83% FLOPs (B10-10-10) still falls behind the standard Transformer that always maintains a full-length token-level representations. This suggests sequential compression could harm the performance when detailed token-level information is critical. On the other hand, compared to the results on SQuAD1.1, F-TFMs perform relatively better on SQuAD2.0,

| Model | CoLA | SST-2 | MRPC | STS-B | QQP | MNLI | QNLI | RTE | WNLI | AVG |
|---|---|---|---|---|---|---|---|---|---|---|
| *Dev set results (single model)* | | | | | | | | | | |
| ROBERTA_Large [4] | 68.0 | 96.4 | -/90.9 | 92.4 | -/92.2 | 90.2 | 94.7 | 86.6 | - | 88.9 |
| XLNet_Large [3] | 69.0 | 97.0 | -/90.8 | 92.5 | -/92.3 | 90.8 | 94.9 | 85.9 | - | 89.2 |
| ELECTRA_Large [5] | 69.1 | 96.9 | -/90.8 | 92.6 | -/92.4 | 90.9 | 95.0 | 88.0 | - | 89.5 |
| B10-10-10H1024 | 72.4 | 96.8 | 93.5/90.9 | 92.1 | 89.8/92.4 | 91.1/- | 95.1 | 89.5 | - | **90.0** |
| B8-8-8H1024 | 71.3 | 96.8 | 93.1/90.7 | 91.7 | 89.8/92.4 | 90.8/- | 94.7 | 89.2 | - | 89.7 |
| ROBERTA_Base [4] | 63.6 | 94.8 | -/90.2 | 91.2 | -/91.9 | 87.6/- | 92.8 | 78.7 | - | 86.4 |
| MPNet_Base [12] | 65.0 | 95.4 | -/91.5 | 90.9 | -/91.9 | 88.5/- | 93.3 | 85.2 | - | 87.7 |
| B6-6-6H768 | 70.1 | 96.3 | 93.2/90.4 | 91.1 | 89.2/92.0 | 89.7/- | 93.7 | 83.4 | - | **88.3** |
| B6-3x2-3x2H768 | 68.5 | 95.6 | 92.5/89.5 | 91.0 | 89.3/92.0 | 89.1/- | 93.0 | 83.4 | - | 87.8 |
| B4-4-4H768 | 68.2 | 95.0 | 92.8/90.2 | 90.3 | 89.0/91.8 | 88.6/- | 92.6 | 79.1 | - | 87.0 |
| *Leaderboard test set results (single task & single model)* | | | | | | | | | | |
| ELECTRA_Large [5] | 68.1 | 96.7 | 89.2/92.0 | 92.1/91.7 | 74.8/90.4 | 90.7/90.2 | 95.5 | 86.1 | 65.1 | 85.2 |
| B10-10-10H1024 | 68.9 | 97.2 | 89.4/92.1 | 91.6/91.3 | 74.3/90.2 | 90.9/90.9 | 95.5 | 86.5 | 65.1 | **85.4** |
| B8-8-8H1024 | 68.3 | 96.9 | 89.2/92.0 | 91.5/91.1 | 73.8/90.1 | 90.7/90.7 | 95.1 | 85.3 | 65.1 | 85.0 |
| ELECTRA_Base [5] | 64.6 | 96.0 | 88.1/91.2 | 91.0/90.2 | 73.2/89.5 | 88.5/88.0 | 93.1 | 75.2 | 65.1 | 82.7 |
| B6-6-6H768 | 68.3 | 96.5 | 89.1/91.9 | 90.6/89.9 | 73.3/89.9 | 89.7/89.4 | 94.0 | 80.4 | 65.1 | **84.0** |
| B6-3x2-3x2H768 | 65.9 | 96.0 | 87.8/91.0 | 90.0/89.6 | 73.3/89.8 | 88.9/88.7 | 93.8 | 79.9 | 65.1 | 83.4 |
| *Leaderboard test set results (multi-task & ensemble)* | | | | | | | | | | |
| ROBERTA_Large [4] | 67.8 | 96.7 | 89.8/92.3 | 92.2/91.9 | 74.3/90.2 | 90.8/90.2 | 95.4 | 88.2 | 89.0 | 88.1 |
| ELECTRA_Large [5] | 71.7 | 97.1 | 90.7/93.1 | 92.9/92.5 | 75.6/90.8 | 91.3/90.8 | 95.8 | 89.8 | 91.8 | 89.4 |
| B10-10-10H1024 | 70.5 | 97.5 | 91.2/93.4 | 92.6/92.3 | 75.4/90.7 | 91.4/91.1 | 95.8 | 90.0 | 94.5 | **89.7** |

Table 3: Comparison with previous methods on the GLUE benchmark under large-scale pretraining.

| Model | RACE | | |
|---|---|---|---|
| | Total | High | Middle |
| ROBERTA_Large [4] | 83.2 | 81.3 | 86.5 |
| XLNet_Large [3] | 85.4 | 84.0 | 88.6 |
| B10-10-10 | **85.7** | **84.4** | **88.8** |
| B8-8-8 | 85.2 | 83.9 | 88.4 |
| ALBERT_Base [6] | 66.0 | - | - |
| MPNet_Base [12] | 72.0 | 76.3 | 70.3 |
| B6-6-6 | **79.7** | **78.2** | **83.4** |
| B6-3x2-3x2 | 78.8 | 77.5 | 82.0 |
| B4-4-4 | 76.2 | 74.6 | 80.0 |

Table 4: RACE test performance comparison.

| Model | SQuAD2.0 | | SQuAD1.1 | |
|---|---|---|---|---|
| | EM | F1 | EM | F1 |
| ROBERTA_Large [4] | 86.5 | 89.4 | 88.9 | 94.6 |
| ELECTRA_Large [5] | **88.0** | **90.6** | **89.7** | **94.9** |
| B10-10-10 | 87.6 | 90.4 | 89.0 | 94.7 |
| B8-8-8 | 87.1 | 89.8 | 88.7 | 94.4 |
| ROBERTA_Base [4] | 80.5 | 83.7 | 84.6 | 91.5 |
| MPNet_Base [21] | 80.5 | 83.3 | 86.8 | 92.5 |
| B6-6-6 | **85.1** | **87.7** | **87.4** | **93.3** |
| B6-3x2-3x2 | 84.2 | 87.0 | 87.0 | 93.0 |
| B4-4-4 | 82.6 | 85.5 | 85.9 | 92.2 |

Table 5: SQuAD dev performance comparison.

which additionally requires the model to make a sequence-level prediction on whether the question is answerable. This again shows the general effectiveness of the F-TFM in sequence-level tasks.

## 4.3 Ablation Study

| ID | Layout | (FLOPs / Params) | Pool-Op | Pool-query-only | Sep [cls] | Rel-Attn | GLUE-AVG |
|---|---|---|---|---|---|---|---|
| (1) | B6-6-6 | (1.00x / 1.00x) | Mean | ✓ | ✓ | ✓ | **83.5** |
| (2) | | | Mean | ✓ | | ✓ | 82.9 |
| (3) | | | Mean | | ✓ | ✓ | 83.0 |
| (4) | | | Mean | ✓ | ✓ | | 81.4 |
| (5) | | | Max | ✓ | ✓ | ✓ | 83.4 |
| (6) | | | Top-Attn | ✓ | ✓ | ✓ | 75.8 |
| (7) | B8-8 | (1.14x / 0.91x) | Mean | ✓ | ✓ | ✓ | 83.4 |
| (8) | B5-5-5-5 | (0.89x / 1.08x) | Mean | ✓ | ✓ | ✓ | 82.9 |

Table 6: Ablation study of F-TFMs with different designs.

Finally, based on the GLUE benchmark, we perform a series of ablation studies on the importance of various designs in F-TFM, including the block layout design, the type of pooling operation, the

pool-query-only technique, maintaining a separate `[cls]` vector and the usage of Transformer-XL parameterization.

- Pooling operation: Including the mean pooling we finally employ in F-TFM, we actually test two types of pooling operations.

  (1) The first type is just the strided mean/max pooling as described in section 2.

  (2) The second type aims to select a subset of "hub" states, which refer to those hidden vectors that are attended most in the previous S-Attn layer and hence likely to carry most critical information about the sequence. Concretely, given the attention map from the previous S-Attn layer, we reduce sum the scores along the number of head and query length dimensions to a score for each position. Then, we simply choose the top 50% of states to achieve the same compression rate. Note that, this type of pooling operation is essentially the same as the important states selection procedure in Power-BERT [20].

- Pool-query-only design

- Separating `[cls]` in the pooling operation

- Block layout design: In our experiments, all models actually utilize a 3-block design. Here, we compare the 3-blocks design with the 2-blocks and the 4-blocks design.

- Relative attention parameterization proposed in Transformer-XL [40]. We compare this parameterization with the learned absolute position embedding as used in the BERT [2].

The ablation results are included in Table 6. To save the computation resources, the size of model hidden states in table 6 is set as 512. From the ablation results, we can make the following observations:

- Comparing pooling different operation ((1), (5), and (6)), we found that the performance of the mean and max pooling operation is similar. But they are significantly better than the idea of utilizing attention score (Top-Attn pooling) to select the "hub" states.

- Comparing (1) with (2) and (3) respectively, we see that the two special designs, i.e. "pool-query-only" and maintaining a separate non-pooled `[cls]` , can both bring a clear improvement to the proposed model.

- Comparing (1) and (4), we find that the relative positional parameterization is key to the performance of the proposed F-TFM. We suspect that the pooling operation could destroy the positional information carried by the absolute position encoding, which is only injected to the model in the input embedding layer. As a result, the higher blocks may not have enough positional information to learn a good enough attention pattern. In comparison, the positional information is injected to each layer under the relative positional attention scheme. Therefore, to achieve good result with F-TFM based on absolute positional embedding, one may inject the absolute positional embedding into each attention layer. Actually, a contemporary application of Transformer to the detection problem in computer vision shows injecting positional embedding into each layer is important [41].

- Finally, we study the influence of block layout design in our framework. With B6-6-6 as the 3-block benchmark, we consider two other layout design with similar FLOPs and number of parameters. Specifically, we consider B8-8 for the 2-block design and B5-5-5-5 for the 4-block design. Comparing the results in (1), (7), and (8), we find that the performance of the 3-block (B6-6-6) design achieves the best performance, which is significantly better than the 4-block design and slightly better than the 2-block design. However, if we further taking the FLOPs/#Params into consideration, it is more clear that the 3-block design is superior. Therefore, in the main paper, we always use the 3-block design.

## 5 Conclusion & Discussion

In this work, under the pretraining-finetuning paradigm, we investigate a largely overlooked dimension of complexity in language processing. With the proposed Funnel-Transformer, we show how sequential resolution can be compressed in a simple form to save computation and how the saved FLOPs can be re-invested in improving the model capacity and hence the performance. Open challenges for future research include the better ways to improve the compression scheme, to optimize the block layout design and to re-invest the saved FLOPs. In addition, combining Funnel-Transformer with model compression techniques like knowledge distillation and quantization would be an important direction towards the enhancement of practical impact.

## Broader Impact

Fundamentally, this work proposed a more efficient architecture from a different dimension of model design. We believe this architecture as well as the code and checkpoints to be released can most benefit the field of language processing, with the potential to benefit other fields involving sequence modeling. However, scientifically, the key idea of compressing the sequence resolution in Funnel-Transformer will not always be a good prior for all problems. In addition, as a common problem for neural models, adversarial examples or attacks could alter the behavior and performance of the proposed model dramatically, as well as outlier examples. Therefore, the model should always be applied with caution in practice.

## Acknowledgments and Disclosure of Funding

This work is supported in part by the National Science Foundation (NSF) under grants IIS-1546329 and the Air Force Research Laboratory under agreement number FA8750-19-2-0200. The U.S. Government is authorized to reproduce and distribute reprints for Governmental purposes notwithstanding any copyright notation thereon. The views and conclusions contained herein are those of the authors and should not be interpreted as necessarily representing the official policies or endorsements, either expressed or implied, of the Air Force Research Laboratory or the U.S. Government.

## Footnotes

[2]Since the corresponding memory complexity is simply $O(T^2 + TD)$, which is always offset by a multiplier $1/D$, we will focus on the computation complexity with the conclusion directly carried through.

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
