[Supplementary Material]

# A    Implementation Optimization

## A.1    Sequence Truncation for Separating `[cls]` trick

As discussed in Section 2.2, to avoid breaking the `[cls]` structure commonly used in pretraining, we do not apply the pooling operation to the `[cls]` and keep the hidden state corresponding to `[cls]` intact. While conceptually simple, a naive implementation could slow down the computation by 15% due to the "irregular" sequence length caused by such an operation. Specifically, assume that sequence length of an input sample is a power of two, i.e., $2^p$, which usually is 512 in the pretraining phase. After one pooling operation with the `[cls]` intact, the length of the pooled sequence becomes $2^{p-1} + 1$, which is not a power of 2 anymore. As a result, it can cause memory misalignment and the waste of paralleled computation power in accelerators, leading to substantial speed loss.

To resolve this issue, we employ a simple strategy to truncate the last token after the pooling. Formally, denoting the pooled hidden state as $\mathbf{h} = \{h_{\texttt{[cls]}}, h_1, \cdots, h_{2^{p-1}}\}$, the truncation can be expressed as

$$\hat{\mathbf{h}} = \mathrm{truncate}(\mathbf{h}) = [h_{\texttt{[cls]}}, h_1, \cdots, h_{2^{p-1}-1}] \tag{6}$$

With this simple trick, we can always keep the sequence length a power of 2, hence avoiding the slowdown caused by maintaining an independent `[cls]` hidden state.

## A.2    Relative Positional Attention Implementation

In this work, we use the relative positional attention parameterization proposed in the Transformer-XL [40]. To facilitate further discussion, we first review the details of this parameterization. Taking the case of single head attention as the example head. Let $T, D$ be the sequence length and hidden dimension respectively. Then, the pre-softmax attention score $A_{ij}$ between a pair of positions $i$ and $j$ consists of two terms:

$$A_{ij} = \underbrace{(W_Q h_i + v)^\top (W_K h_j)}_{\text{content term}} + \underbrace{(W_Q h_i + u)^\top (W_R r_{i-j})}_{\text{position term}}. \tag{7}$$

where $v, u \in \mathbb{R}^D$ are two trainable bias vectors, $W_Q, W_K, W_R \in \mathbb{R}^{D \times D}$ are three trainable projection matrices, and $r_{i-j} \in \mathbb{R}^D$ is the sinusoidal positional encoding that represents the relative distance $i - j$ between the two positions.

To compute the entire attention score matrix $\mathbf{A}$, the content term can easily be obtained via two head projections and an *outer product* of complexity $O(TD^2 + T^2 D)$:

$$\mathbf{A}^{\text{content}} = (\mathbf{H}W_Q + v)(\mathbf{H}W_K)^\top,$$

where $\mathbf{H} = [h_1, \cdots, h_T] \in \mathbb{R}^{T \times D}$ collects all hidden states into a matrix. However, we cannot compute the position term in the same way as each $A_{ij}^{\text{position}}$ corresponds to a different $r_{i-j}$. Hence, a naive solution will be stacking $T^2$ pairs of position encodings into a tensor $\hat{\mathbf{R}} \in \mathbb{R}^{T \times T \times D}$ where $\hat{\mathbf{R}}_{ij} = r_{i-j}$, and then perform the following tensor product:

$$\mathbf{A}^{\text{position}} = \texttt{einsum("id,ijd->ij"}, \mathbf{H}W_Q + u, \hat{\mathbf{R}}W_R).$$

Note that the head projection $\mathbf{R}W_K$ now has a complexity of $O(T^2 D^2)$ and a memory footprint of $O(T^2 D)$, dominating all other computations.

### A.2.1    Standard Solution: Gather / Shift

To resolve the computation burden above, a common technique is to instead collect a matrix $\mathbf{R} \in \mathbb{R}^{2T-1 \times D}$, where

$$\mathbf{R} = [r_{T-1}, \ldots, r_0, \cdots, r_{1-T}]$$

which includes all possible position encodings arranged from the maximum possible distance value $T - 1$ to the minimum one $1 - T$. Note that the full $\hat{\mathbf{R}}$ can be formed by gathering specific elements from $\mathbf{R}$ with an index matrix $\mathbf{I}$ of shape $[T \times T]$, i.e.,

$$\hat{\mathbf{R}} = \mathrm{gather}(\mathbf{R}, \mathbf{I}), \quad I_{ij} = T + i - j.$$

Mathematically, this is equivalent to using a permutation tensor $\mathbf{P} \in \mathbb{R}^{T \times T \times 2T-1}$ to multiply $\mathbf{R}$, i.e., $\hat{\mathbf{R}} = \mathbf{P}\mathbf{R}$, where $\mathbf{P}_{ij} \in \mathbb{R}^{2T-1}$ is a one-hot vector used to select/gather a single position of $\mathbf{R}$. As

the attention score computation only involves linear operations, we can rearrange the computation of the position term as follows

$$\mathbf{A}^{\text{position}} = \texttt{einsum}(\texttt{"id,ijd->ij"}, \mathbf{H}W_Q + u, (\mathbf{PR})W_R)$$

$$= \texttt{einsum}\Big(\texttt{"ijk,jk->ij"}, \mathbf{P}, \big[(\mathbf{H}W_Q + v)(\mathbf{R}W_R)^\top\big]\Big)$$

$$= \texttt{gather}\Big((\mathbf{H}W_Q + v)(\mathbf{R}W_R)^\top, \mathbf{I}\Big)$$

Note that, assuming gathering $T^2$ elements only has a complexity of $O(T^2)$, which is true for CPU/GPU, this trick reduces the computation complexity back to $O(2TD^2 + 2T^2D)$. In practice, the gather operation can be implemented via a smart reshape operation, that is even cheaper.

### A.2.2 Optimization for TPU: factorized relative positional attention

However, on TPUs, the assumption that gathering $T^2$ elements only has a complexity of $O(T^2)$ does not hold. Instead, we found that such a gather operation is dramatically slower on TPU. Hence, we here consider another implementation which is significantly faster on TPU.

Firstly, let's rewrite the position term as follows

$$A_{ij}^{\text{position}} = (W_Q h_i + u)^\top (W_R r_{i-j})$$

$$= \Big[ \underbrace{W_R^\top (W_Q h_i + u)}_{q_i} \Big]^\top r_{i-j}$$

$$= q_i^\top r_{i-j}. \tag{8}$$

For easier derivation, we have introduced a notation of $q_i$. Then, recall the $r_{i-j}$ is the sinusoidal encoding that consists of the sine and the cosine components $r_{i-j} = \texttt{cat}(\sin_{i-j}, \cos_{i-j})$, where

$$\sin_t = \Big[\sin\Big(t/10000^{2/D}\Big), \sin\Big(t/10000^{4/D}\Big), \cdots, \sin\Big(t/10000^{D/D}\Big)\Big] \in \mathbb{R}^{D/2},$$

$$\cos_t = \Big[\cos\Big(t/10000^{2/D}\Big), \cos\Big(t/10000^{4/D}\Big), \cdots, \cos\Big(t/10000^{D/D}\Big)\Big] \in \mathbb{R}^{D/2}.$$

Hence, we similarly divide $q_i$ defined above into two parts, i.e.,

$$q_i = \texttt{cat}(q_i^{\text{sin}}, q_i^{\text{cos}}).$$

Given the definitions, we can further break Eqn. (8) into two terms:

$$A_{ij}^{\text{position}} = q_i^\top r_{i-j} = q_i^{\text{sin}\top} \sin_{i-j} + q_i^{\text{cos}\top} \cos_{i-j}.$$

Now, using the trigonometric identities $\sin(a-b) = \sin(a)\cos(b) - \cos(a)\sin(b)$ and $\cos(a-b) = \cos(a)\cos(b) + \sin(a)\sin(b)$, the two terms can be respectively reformulated into

$$q_i^{\text{sin}\top} \sin_{i-j} = q_i^{\text{sin}\top} [\sin_i \odot \cos_j - \cos_i \odot \sin_j]$$

$$= q_i^{\text{sin}\top} (\sin_i \odot \cos_j) - q_i^{\text{sin}\top} (\cos_i \odot \sin_j)$$

$$= \big[q_i^{\text{sin}} \odot \sin_i\big]^\top \cos_j + \big[q_i^{\text{sin}} \odot (-\cos_i)\big]^\top \sin_j$$

and

$$q_i^{\text{cos}\top} \cos_{i-j} = q_i^{\text{cos}\top} [\cos_i \odot \cos_j + \sin_i \odot \sin_j]$$

$$= q_i^{\text{cos}\top} (\cos_i \odot \cos_j) + q_i^{\text{cos}\top} (\sin_i \odot \sin_j)$$

$$= \big[q_i^{\text{cos}} \odot \cos_i\big]^\top \cos_j + \big[q_i^{\text{cos}} \odot \sin_i\big]^\top \sin_j$$

Hence, combining these two parts together, it follows that

$$q_i^\top r_{i-j} = {q_i^{\text{sin}}}^\top \sin_{i-j} + {q_i^{\text{cos}}}^\top \cos_{i-j}$$

$$= \left[q_i^{\text{sin}} \odot \sin_i\right]^\top \cos_j + \left[q_i^{\text{sin}} \odot (-\cos_i)\right]^\top \sin_j + \left[q_i^{\text{cos}} \odot \cos_i\right]^\top \cos_j + \left[q_i^{\text{cos}} \odot \sin_i\right]^\top \sin_j$$

$$= \left\{\left[q_i^{\text{sin}} \odot \sin_i\right]^\top \cos_j + \left[q_i^{\text{cos}} \odot \cos_i\right]^\top \cos_j\right\} + \left\{\left[q_i^{\text{sin}} \odot (-\cos_i)\right]^\top \sin_j + \left[q_i^{\text{cos}} \odot \sin_i\right]^\top \sin_j\right\}$$

$$= \left[\underbrace{\texttt{cat}(q_i^{\text{sin}}, q_i^{\text{cos}})}_{=q_i} \odot \underbrace{\texttt{cat}(\sin_i, \cos_i)}_{:=\phi_i}\right]^\top \underbrace{\texttt{cat}(\cos_j, \cos_j)}_{:=\psi_j}$$

$$+ \left[\underbrace{\texttt{cat}(q_i^{\text{sin}}, q_i^{\text{cos}})}_{=q_i} \odot \underbrace{\texttt{cat}(-\cos_i, \sin_i)}_{:=\pi_i}\right]^\top \underbrace{\texttt{cat}(\sin_j, \sin_j)}_{:=\omega_j}$$

$$= \left[q_i \odot \phi_i\right]^\top \psi_j + \left[q_i \odot \pi_i\right]^\top \omega_j,$$

where $\phi_i, \psi_j, \pi_i, \omega_j$ above are simply 4 positional encodings formed by concatenating the cosine and sine vectors of the corresponding $i$ and $j$ in different ways. Note that, each term of the last line has a *factorized* form that can be computed via an *outer product*, just like the standard content term. Therefore, by stacking $\phi_i, \psi_j, \pi_i, \omega_j$ of all positions (i.e. $i = 1, \ldots, T$ and $j = 1, \ldots, T$) into the corresponding $\mathbf{\Phi}, \mathbf{\Psi}, \mathbf{\Pi}, \mathbf{\Omega} \in \mathbb{R}^{T \times D}$ respectively, the full position term can be expressed in a simple form

$$\mathbf{A}^{\text{position}} = \left\{\left[(\mathbf{H}W_Q + u)W_R^\top\right] \odot \mathbf{\Phi}\right\}\mathbf{\Psi}^\top + \left\{\left[(\mathbf{H}W_Q + u)W_R^\top\right] \odot \mathbf{\Pi}\right\}\mathbf{\Omega}^\top$$

which leads to the complexity of $O(2TD^2 + 4T^2D)$, which is comparable to the content term.

## A.3 Potential Model Extensions

In this section, we discuss some potential model extensions of Funnel-Transformer. As described in section 2, Funnel-Transformer can be divided into an encoder with a compression functionality and a decoder that recovers the full-length token-level representations. To further extend the proposed model, first note that the encoder-decoder framework can be formulated into a more general form:

$$\mathbf{h}_{\text{enc}} = \text{Encoder}(\mathbf{x}_{\text{enc}}),$$
$$\mathbf{h}_{\text{dec}} = \text{Decoder}(\mathbf{h}_{\text{enc}}, \mathbf{x}_{\text{dec}}),$$

where $\mathbf{x}_{\text{enc}}$ and $\mathbf{x}_{\text{dec}}$ are the encoder input sequence and the *optional* and *problem-specific* decoder input, respectively. The goal of encoder is to compressing the input sequence $\mathbf{x}_{\text{enc}}$ into the hidden representations $\mathbf{h}_{\text{enc}}$ with a reduced length. Then, conditioned on the decoder input $\mathbf{h}_{\text{enc}}$ if any, the decoder will extract relevant information/representations from $\mathbf{h}_{\text{enc}}$ to solve the specific NLP problem at hand. Next, we will how the general form of Funnel-Transformer can be instantiated into specific forms to solve corresponding NLP problems.

**Sequence-level prediction**   This is essentially the case we consider in most of our experiments where we want to obtain a vectorial representation of the input sequence such as text classification. In this case, we don't really need the decoder $\mathbf{x}_{\text{dec}}$ (i.e. $\mathbf{x}_{\text{dec}} = \varnothing$) and the decoder simply extracts the hidden representation corresponding to the [cls] token from $\mathbf{h}_{\text{enc}}$ and feeds it into the task-specific structure (e.g. classifier).

**Token-level prediction**   In the token-level prediction tasks such as the MLM pretraining, SQuAD and sequence labeling, we need a decoder to recover the token-level representations from the compressed sequence $\mathbf{h}_{\text{enc}}$. In many cases, $\mathbf{x}_{\text{dec}}$ could simply be the original sequence or a token-level hidden representation of it to provide fine grained low-level information of each token and hence ease the optimization. In this paper, we utilize the last-layer hidden states of the 1st block (before the first pooling operation) as the additional decoder input.

But for problems that utilize additional input signals, such as the permutation order used for permuted language modeling in XLNet [3]. This additional information can be injected into Funnel-Transformer via the decoder input $\mathbf{x}_{\text{dec}}$ to (approximately) recover some more complex control of attention mechanism.

**Sequence-to-sequence problems**   Another important category of NLP task is sequence-to-sequence problems, including machine translation, text summarization, and dialog generation, whose state-of-the-art solution is the conventional encoder-decoder framework. Hence, Funnel-Transformer naturally fits these tasks, where the decoder input $x_{dec}$ corresponds to the target text sequence and the encoder input $x_{enc}$ the source text sequence. This way, the key difference compared to conventional models is the source side compression Funnel-Transformer provides.

Overall, we summarize some potential directions to extend Funnel-Transformer presented in section 2.2 to NLP problems. Finally, although we focus on discussion on the NLP tasks in this paper, Funnel-Transformer could be applied to any tasks dealing with sequential data, such as time series and video stream analysis.

# B   Experiment Setting and Hyper-parameters

## B.1   Preprocessing & Tokenization

For all experiments conducted in this work, we simply adapt the "uncased" word piece model originally used by BERT [2], where the vocabulary size is about 30K. Other than lower case and the default preprocessing included in the word piece tokenizer, the only additional preprocessing we perform is to remove some http symbols (e.g. <b>) in the 7 text classification tasks.

## B.2   Pretraining

| Hparam | Base Scale | Large Scale |
|---|---|---|
| Hidden dropout | 0.1 | |
| GeLU dropout | 0.0 | |
| Attention dropout | 0.1 | |
| Max sequence length | 512 | |
| Batch size | 256 | 8192 |
| Learning rate | 1e-4 | 2e-4 |
| Number of steps | 1M | 500K |
| Warmup steps | 10K | 30K |
| Optimizer | Adam Weight Decay | |
| Learning rate decay | Linear | |
| Adam epsilon | 1e-6 | |
| Weight decay | 0.01 | |

Table 7: Hyper-parameters for pretraining.

The hyper-parameters used for the two different pretraining settings are summarized in Table 7. One exception is the learning rate used for B10-10-10H1024 at the base scale. Specifically, we find the training can be unstable when the depth goes beyond 24 layers (in the case of B10-10-10H1024) at base scale, especially for the MLM objective. Hence, we reduce the learning to 8e-5 for the B10-10-10H1024 F-TFM during base-scale pretraining. This has a side effect of a slower training pace and potentially a slightly worse finetuning performance. However, we does not observe such instability when the batch size is increased such as in the large-scale setting.

For ELECTRA, there are two additional important hyper-parameters, i.e., the discriminator loss coefficient and the relative size multiplier of the generator. In this work, we does not tune these two hyper-parameters at all and simply use the numbers from the original paper, i.e., the discriminator loss coefficient of 50 and size multiplier of 1/4 for all architectures trained with ELECTRA. In addition, in ELECTRA training, whenever F-TFM is used as the discriminator, the generator also uses the F-TFM.

In additional, in the all experiments, we only annotate the size of hidden states the rest of model sizes can be derived from on it:

- The embedding size = hidden size
- The size of inner states of P-FFN is "$4 \times$ hidden size".
- The attention head dimension is always $64$.
- The number of attention heads is "hidden size$/64$".

Finally, another important element in pretraining is the mask sampling strategy. For MLM training, following previous work, we always complete word span (up to 5 complete words) sampling. However, for ELECTRA training, we notice a weird phenomenon that under the base-scale setting, the performance of both the Transformer and the F-TFM drops significantly if we use word span sampling rather than the single-token sampling. On the other hand, under the large-scale setting, using word span sampling works fine. Hence, we use single-token sampling for base-scale ELECTRA training, and word span sampling for large-scale ELECTRA training.

## B.3 Finetuning

| Hparam | RTE | MRPC | STS-B | CoLA | SST-2 | QNLI | MNLI | QQP |
|---|---|---|---|---|---|---|---|---|
| Hidden dropout | | | | 0.1 | | | | |
| GeLU dropout | | | | 0.0 | | | | |
| Attention dropout | | | | 0.1 | | | | |
| Max sequence length | | | | 128 | | | | |
| Batch size | 16 | 16 | 16 | 16 | 32 | 32 | 64 | 64 |
| Number of epochs | 10 | 10 | 10 | 10 | 5 | 3 | 3 | 5 |
| Learning rate decay | | | | Linear | | | | |
| Weight decay | | | | 0.01 | | | | |
| Warmup proportion | | | | 0.1 | | | | |
| Adam epsilon | | | | 1e-6 | | | | |
| **Hparam** | **IMDB** | **AG** | **DBpedia** | **Yelp-2** | **Yelp-5** | **Amazon-2** | **Amazon-5** | |
| Hidden dropout | | | | 0.1 | | | | |
| GeLU dropout | | | | 0.0 | | | | |
| Attention dropout | | | | 0.1 | | | | |
| Max sequence length | 512 | 128 | 128 | 512 | 512 | 512 | 512 | |
| Batch size | 32 | 32 | 64 | 128 | 128 | 128 | 128 | |
| Number of epochs | 5 | 3 | 3 | 3 | 3 | 3 | 3 | |
| Learning rate decay | | | | Linear | | | | |
| Weight decay | | | | 0.01 | | | | |
| Warmup proportion | | | | 0.1 | | | | |
| Adam epsilon | | | | 1e-6 | | | | |

Table 8: Hyper-parameters for finetuning on the GLUE benchmark and 7 text classification datasets.

For all the finetuning experiments, we essentially inherit the hyper-parameters used by XLNet [3]. All the performance numbers reported are obtained on TPUs with TensorFlow 2.2.

### B.3.1 GLUE & Text Classification

For GLUE and text classification datasets, we first fix the values of most hyper-parameters shown in Table 8. Then, we only search the learning rates from the set [1e-5, 2e-5, 3e-5], and choose the best one according to the validation set.

Following previous work [3, 4, 5], all GLUE performances correspond to the median result of 5 runs from different random seeds in the base setting and 15 runs in the large setting, respectively.

For the text classification, the base-scale results are the median performance among 5 runs with different random seeds. However, for the large-scale experiments, to be compatible with previous work [42, 3], the results are the best performance among 5 random runs.

### B.3.2 Reading Comprehension

Again, following XLNet [3], the hyper-parameters used for finetuning on the RACE and SQuAD datasets are summarized in Table 9. "Layer-wise decay" means exponentially decaying the learning rates of individual layers in a top-down manner. For example, suppose the 24-th layer uses a learning rate $l$, and the Layer-wise decay rate is $\alpha$, then the learning rate of layer $m$ is $l\alpha^{24-m}$. In addition, for the two versions of SQuAD, we simply reuse the model trained on SQuAD v2.0 when evaluated on SQuAD v1.1.

| Hparam | RACE | SQuAD |
|---|---|---|
| Dropout | 0.1 | |
| Attention dropout | 0.1 | |
| Max sequence length | 512 | 512 |
| Training epochs/steps | 5 epochs | 8000 steps |
| Warmup proportion/steps | 0.1 | 1000 steps |
| Batch size | [16, 32] | 48 |
| Learning rate | [1e-5, 2e-5] | 3e-5 |
| Learning rate decay | linear | |
| Weight decay | 0.01 | |
| Adam epsilon | 1e-6 | |
| Layer-wise lr decay | 1.0 | 0.75 |

Table 9: Hyper-parameters for RACE and SQuAD.

# C  Additional Experimental Results

## C.1  Text Classification at Large Scale

| Model | IMDB | AG | DBpedia | Yelp-2 | Yelp-5 | Amazon-2 | Amazon-5 |
|---|---|---|---|---|---|---|---|
| BERT-Large | 4.51 | - | 0.64 | 1.89 | 29.32 | 2.63 | 34.17 |
| ROBERTA-Large | 3.50 | - | - | - | - | - | - |
| XLNet-Large | **3.20** | **4.45** | 0.64 | 1.37 | **27.05** | 2.11 | 31.67 |
| B10-10-10H1024 | 3.36 | 4.66 | **0.60** | **1.33** | 27.14 | **2.10** | **31.64** |
| B8-8-8H1024 | 3.42 | 4.96 | 0.63 | 1.39 | 27.20 | 2.14 | 31.74 |
| MPNet | 4.40 | - | - | - | - | - | - |
| B6-6-6H768 | **3.72** | **5.00** | **0.64** | **1.50** | **27.73** | **2.27** | **32.11** |
| B6-3x2-3x2H768 | 3.82 | 5.12 | 0.64 | 1.58 | 27.96 | 2.32 | 32.23 |
| B4-4-4H768 | 4.12 | 5.09 | 0.67 | 1.70 | 28.40 | 2.35 | 32.46 |

Table 10: Text classification performance comparison under the large-scale pretraining.

Table 10 includes the performance comparison on 7 text classification tasks under the large-scale training setting. Similar to the GLUE benchmark results, compared with the previous result based on Transformer, with fewer FLOPs, the proposed F-TFM achieves comparable results.

## C.2  Training Cost Comparison

In this section, we test the pretraining and finetuning speed of the F-TFM in comparison to the standard Transformer on the TPU and GPU platform. For the pretraining speed evaluation, we test F-TFM on TPU v3-16 (16 cores x 16Gb) with TensorFlow. For the finetuning speed evaluation, we test F-TFM on TPU v2-8 (8 cores x 8Gb) with TensorFlow and on Nvidia-V100 (16Gb) GPU with the PyTorch. The TensorFlow version is 2.2.0, and the PyTorch version is 1.5.0. For the GPU experiments, we use an 8-GPU node on the Google Cloud Platform. All running speeds are reported with the FP16 optimizer. In the PyTorch implementation, we use "O2" options of AMP manager in the apex[3] package to handle the FP16 optimization. For finetuning, we consider three different sequence lengths, namely 128, 256 and 512. For pretraining, we only consider the sequence length 512. In each case, we choose the maximum possible batch size allowed by the memory size of the device(s). We measure the actual model *running time* by performing 1000 steps gradient descent with random input sequences with the fixed length.

Firstly, we compare the model speed in the finetuning stage. Note that the decoder is not used in this setting. Table 11 and 12 summarize the finetuning running time comparison on GPUs and TPUs, respectively.

- In the base model (L12H768) group, we observe that the speed of B6-6-6H768 is similar or faster than the base Transformer model, despite the fact that B6-6-6 is deeper, has more parameters.

| Sequence length | 128 | | | 256 | | | 512 | | |
|---|---|---|---|---|---|---|---|---|---|
| Metrics | Run time | | Mem | Run time | | Mem | Run time | Mem | GLUE |
| | 1 GPU | 8 GPUs | | 1 GPU | 8 GPUs | | 8 GPUs | | |
| Batch size / GPU | 64 | | | 32 | | | 16 | | |
| L12H768 | 1.00x | 1.00x | 9.2G | 1.00x | 1.00x | 11.0G | 1.00x | 14.3G | 84.40 |
| B6-6-6 | 0.97x | 0.99x | 9.1G | 0.95x | 0.97x | 10.3G | 0.94x | 12.5G | 85.37 |
| B6-3x2-3x2 | 0.93x | 0.93x | 8.4G | 0.91x | 0.92x | 9.5G | 0.90x | 11.8G | 84.78 |
| B4-4-4 | 0.67x | 0.67x | 6.6G | 0.65x | 0.66x | 7.5G | 0.64x | 9.0G | 83.99 |
| Batch size / GPU | 32 | | | 12 | | | 4 | | |
| L24H1024 | 1.00x | 1.00x | 14.8G | 1.00x | 1.00x | 14.4G | 1.00x | 13.9G | 86.62 |
| B10-10-10 | 0.87x | 0.92x | 14.0G | 0.90x | 0.93x | 13.0G | 0.96x | 12.7G | 87.03 |
| B8-8-8 | 0.70x | 0.73x | 11.6G | 0.73x | 0.75x | 10.8G | 0.78x | 10.5G | 86.70 |

Table 11: Running time and memory consumption comparison between F-TFMs and the standard Transformer on the GPU. In each model group, the standard Transformer (first model) is used as the benchmark for the rest of F-TFM models. Note that, given the same batch size per GPU, the memory consumption is roughly the same for 1 GPU and 8 GPUs.

| Sequence length | 128 | 256 | 512 | |
|---|---|---|---|---|
| Metrics | Run time on 8 TPU cores (TPUv2-8) | | | GLUE |
| Batch size / TPU core | 64 | 32 | 16 | |
| L12H768 | 1.00x | 1.00x | 1.00x | 84.40 |
| B6-6-6 | 0.99x | 0.88x | 0.81x | 85.37 |
| B6-3x2-3x2 | 0.97x | 0.87x | 0.77x | 84.78 |
| B4-4-4 | 0.69x | 0.62x | 0.55x | 83.99 |
| Batch size / TPU core | 16 | 8 | 4 | |
| L24H1024 | 1.00x | 1.00x | 1.00x | 86.62 |
| B10-10-10 | 0.89x | 0.81x | 0.73x | 87.03 |
| B8-8-8 | 0.66x | 0.60x | 0.56x | 86.70 |

Table 12: Running time between F-TFMs and the standard Transformer on the TPU v2-8. In each model group, the standard Transformer (first model) is used as the benchmark for the rest of F-TFM models.

Moreover, B6-6-6H768 achieves better results compared with the base Transformer model. The similar conclusion applies to the B6-3x2-3x2 model, which has the same amount of parameters as the base model. The B4-4-4 model, which has the same depth and model parameters as the base model, is able to provide 30%-50% speedup without losing too much performance.

- In the large model (L24H1024) group, the conclusion is similar. The speed of the larger model B10-10-10 is almost the same as the large model, and the speed of B8-8-8 is significantly faster than the large model. In addition, when sequence length equals 512, the acceleration of F-TFM on the TPU is more obvious than the GPU.

- In the both groups, all the tested F-TFM variants have smaller memory footprint compared with the standard TFM models, showing the memory efficiency of F-TFM.

Next, we compare the model speed during pretraining under the MLM objective in table 13, which has an additional cost due to the decoder. The results show that the proposed method can still substantially improve the pretraining speed compared to the standard Transformer, though the speed gain is slightly smaller than the finetuning stage. In summary, this study demonstrates that the proposed method is more efficient in both the finetuning and pretraining stages in modern parallel computing platforms.

| Sequence Length | 512 | |
|---|---|---|
| | Running Time | FLOPs |
| #TPU cores / Total bsz | 16 / 512 | |
| L12H768 | 1.00x | 1.00x |
| B6-6-6H768D2 | 0.99x | 1.04x |
| B6-3x2-3x2H768D2 | 0.97x | 1.04x |
| B4-4-4H768D2 | 0.79x | 0.75x |
| #TPU cores / Total bsz | 16 / 128 | |
| L24H1024 | 1.00x | 1.00x |
| B10-10-10H1024D2 | 0.83x | 0.81x |
| B8-8-8H1024D2 | 0.71x | 0.66x |

Table 13: TPU pretraining speed comparison. The suffix "D2" means that the F-TFM model has 2 decoder layers.

## Footnotes

[3] https://github.com/NVIDIA/apex