[Reviews · NeurIPS 2020]

Review 1

Summary and Contributions: This paper proposes a method that extends the Transformer in terms of computational efficiency. The main idea of the proposed method is to shrink the network in the sequence (or sentence) length direction. The experiments on several text classification tasks show that the performance of the proposed method achieved or outperforms the SOTA results.

Strengths: 1, The experimental results on text classification and some other text processing tasks are fairly well. 2, The idea to reduce the computational cost in the sentence length wide is novel and possibly has a large impact to the community.

Weaknesses: 1, The idea of shrinking the network is somewhat not novel in the community since convolutional neural networks widely used in the image recognition tasks gradually merge and reduce the spatial resolution as the neural network goes deeper. ===> after author response: In the rebuttal: “Indeed, shrinking spatial dimension is not an entirely new idea. However, despite its prevalence in ConvNets, this idea hasn’t not been successfully applied in NLP, especially under the context of pretraining where (1) the model capacity and scalability are critical, but (2) reducing the length could harm the capacity. Thus, this work can be seen a proof of concept such a trade-off can be beneficial even under the context of pretraining.” The idea of gradually shrinking the dimensions is well-known in the community so that I still think the idea of the proposed method is not novel nor innovative since the higher-level concept seems identical to the community knowledge. From this sense, this part is the evident weakness of this paper. However, this paper has several strong points such as a length reduction method and comprehensive experiments. Overall, I think that this paper has enough contributions to be accepted to the conference.

Correctness: 1, The main claim of this paper seems valid and correct. 2, The computational efficiency was evaluated by FLOPs. The FLOPs do not rely on the actual runtime. Therefore, the discussions of efficiency are not very convincing and might mislead the readers to the wrong conclusion. ===> after author response: In the rebuttal: “We agree FLOPs don’t tell the whole story. Therefore, in addition to FLOPs, we indeed compare the exact running time in the Appendix C.3, which also shows the efficiency gain of the proposed model.” Indeed, I found the statement in L273, “Hence, for completeness, we show the speedup provided by the F-TFM in terms of actual running time in Appendix C.3.” However, I believe, in the sprits of the conference paper and the reviewing guideline, the reviewer should review the contents of the main body. Therefore, I do not consider the results of the actual runtime as the strength of the paper.

Clarity: The paper is basically well-written, and the most part is understandable.

Relation to Prior Work: There are tons of related studies to this paper since to propose improved architectures of the Transformers is one of the recent trends in the ML/NLP research community. This paper tries to enumerate many related papers, and empirically compares several important related methods in the experiment section.

Reproducibility: Yes

Additional Feedback:


Review 2

Summary and Contributions: I have read the authors' rebuttal and other reviewers' comments. I will keep my score since the authors have proposed some new ideas like (length reduction, pooling operation, encoder-decoder) in NLU tasks and addressed my concerns. ========================= This paper introduces a novel pre-trained approach, which incorporates sequence compression and parameter sharing into the Transformer model, and thus reduces the computation cost. Specifically, it conducts a pooling operation over the hidden states. In order to maintain the representation of each token, it further borrows ideas from image generation or super-resolution, which uses a decoder to recover the compressed sequence by performing the up-sampling operation. Benefits from such design, it can save computations and thus allow the model to use deeper structure and achieve advanced performance.

Strengths: 1. This paper introduces a novel idea which use sequence compressing to save computation cost to improve efficiency. 2. In order to recover token-level representation, it borrows the idea of image generation or super-resolution, which performs a decoder for sequence up-sampling for MLM learning or sequence labeling. 3. Experimental results on different pre-training objectives (MLM and Electra) validate the effectiveness of funnel-transformer. 4. It also discusses the potential directions of Funnel-transformer to other NLP problems.

Weaknesses: 1. The idea of this method is interesting, but seems a little complex. In addition, just like authors say, it is difficult to decide the optimal block layout when given the depth-length trade-off.

Correctness: Yes, I confirm the claims and the proposed method are correct.

Clarity: Yes, this paper is well-organized.

Relation to Prior Work: Yes, this paper has discussed the difference between the previous works in natural language processing and computer vision domains.

Reproducibility: Yes

Additional Feedback: 1. Accoding to the original paper, authors will remove decoder during the fine-tuning stage. So have you tried to fine-tune the encoder plus decoder for classification task?


Review 3

Summary and Contributions: This paper proposes an efficient Funnel-Transformer architecture for sequential modeling where the hidden states of the sequences are gradually compressed into shorter ones. Reducing the sequence lengths saves computational budgets which can be redirected towards building a wider/deeper model. Empirical results on natural language pre-training show the proposed model out-perform the standard Transformer given comparable computation budgets.

Strengths: - As far as I know, it is the first time that reducing the sequence lengths for efficient sequential modeling is introduced into language pre-training, although similar technique has been explored in image processing community. - Exploiting the self-attention mechanism for sequence pooling is also interesting. - The experimental result is strong, showing good results against strong baselines. Overall, I like this paper and find it interesting.

Weaknesses: - The experiment only covers natural language understanding tasks. I would like to see how the model performs on natural language generation tasks such as summarization.

Correctness: Yes.

Clarity: The paper is well written and easy to follow.

Relation to Prior Work: The paper clearly discussed its related work in both natural language and image processing community.

Reproducibility: Yes

Additional Feedback: How does the technique scale to longer sequences?


Review 4

Summary and Contributions: This paper proposes a more efficient (i.e., more effective and more cost-effective) alternative to transformer called Funnel-Transformer, which gradually reduces the sequential resolution (i.e. length) of the hidden representation in self-attention models. In addition, the authors use sufficient experiments to demonstrate that their model achieves an improved performance over the standard Transformer on a wide variety of sequence-level prediction tasks, including text classification, language understanding, and reading comprehension.

Strengths: 1. The proposed architecture is simple and the description is also clear. 2. This paper has proved the overall effectiveness of their method on a large number of tasks.

Weaknesses: 1. In the aspect of theory, some key problems have not been well explained and proved: (1) One goal of this paper is to save the training cost of transformer. But the standard transformer can handle the whole text in parallel, why can length reduction effectively improve training speed of transformer? In addition, the proposed models have more parameters and hidden layers than the standard BERT (in Table 1 and 2), why their training speeds are faster than standard BERT (in Table 11) ? For training time, the authors only measure the actual model running time by performing 1000 steps gradient descent with random input sequences with the fixed length. But I’m more curious about total training time of proposed models and standard BERT model, because more parameters and hidden layers often means more epochs for training. (2) In experiments, the proposed Funnel-Transformer steadily surpasses the standard transformer in effect. But it is not clear where this effect improvement comes from, length reduction or increasing hidden layers? Is “trading sequential resolution for more layers” (line 248) means that length reduction will impair the effect and the effect improvement of Funnel-Transformer is caused by increasing layers? If so, why not directly increase the layers of standard transformer to improve the effect and use the trick of parameter sharing to prevent the training cost from increasing? What is the significance of length reduction? I think you should clearly explain and prove this basic problem. 2. In the aspect of experiments, I think that this paper lacks in-depth exploration experiments. The experiments in the text are all used to prove the overall performance of various Funnel-Transformer based models. In fact, I think Section 4.1 or Section 4.2 alone is enough to verify the effectiveness of Funnel-Transformer. I strongly recommend giving some in-depth experiments to explore the problems I mentioned above. 3. Btw, length reduction is an importance improvement of this paper so I think there should be more theoretical analysis and experiments on length reduction to give readers a better understanding of your work.

Correctness: Uncertain, the specific reasons are as above.

Clarity: Yes, it is easy to follow.

Relation to Prior Work: The author does not mention.

Reproducibility: No

Additional Feedback:

[Author Response · NeurIPS 2020]

First of all, we would like to thank all reviewers for the feedback and questions.

**=== Reviewer #1 ===**

**Q**: "The idea of shrinking the network is somewhat not novel ..."

**A**: Indeed, shrinking spatial dimension is not an entirely new idea. However, despite its prevalence in ConvNets, this idea hasn't not been successfully applied in NLP, especially under the context of pretraining where (1) the model capacity and scalability are critical, but (2) reducing the length could harm the capacity. Thus, this work can be seen a proof of concept such a trade-off can be beneficial even under the context of pretraining.

**Q**: "The computational efficiency was evaluated by FLOPs."

**A**: We agree FLOPs don't tell the whole story. Therefore, in addition to FLOPs, we indeed compare the exact running time in the Appendix C.3, which also shows the efficiency gain of the proposed model.

**=== Reviewer #2 ====**

**Q**: "The idea of this method is interesting, but seems a little complex... it is difficult to decide the optimal block layout"

**A**: This is indeed a difficulty future work needs to consider. Also, the optimal layout could be problem dependent. Hence, a more elegant solution in our opinion should enable some simple layout finetuning mechanism "after pretraining".

**Q**: "have you tried to fine-tune the encoder plus decoder for classification task?"

**A**: Yes. We tried to finetune the encoder + decoder version on the GLUE benchmark with B6-6-6H768 pretrained in large-scale setting. Out of 8 tasks, using the decoder only improves the performances on 4 tasks, leading to about 0.3 points gains on average compared to the version without decoder, which is not that significant considering the cost.

**==== Reviewer #3 ====**

**Q**: "I would like to see how the model performs on natural language generation tasks":

**A**: The reason why we focus mostly on language understanding tasks is that they are arguably more influenced by the success of language pretraining, which requires significant computation. But we do agree that how to apply this idea (enabling operations on the sequence length) to generation tasks is definitely a very interesting future direction.

**Q**: "How does the technique scale to longer sequences?"

**A**: We tried to pretrain a proposed model with sequence length $T = 1024$ and $D = 1024$. For relative pretraining loss and speed, we observe very similar patterns with $T = 512$ and $D = 1024$. But due to the much longer total time, we didn't finish the pretraining nor finetuning. In our opinion, "sequence compression" is a necessary component for handling super-long sequences, just like how humans handle such cases.

**==== Reviewer #4 ====**

**Q**: "Why are the training speeds of Funnel-Transformer faster than standard Transformer"

**A**: Firstly, the complexity of a standard Transformer layer to process of sequence of length $T$ and hidden dimension $D$ is $O(T^2D + TD^2)$. Hence, by reducing the length from $T$ to a shorter one $T' = T/2^m (m \geq 1)$, the FLOPs needed are also reduced accordingly. Secondly, although computation can be done in parallel *in theory*, current computational device still cannot finish all parallelizable operations *in a single clock*. In practice, these operations still need to be done sequentially. Therefore, reducing FLOPs requirement of the model improves the running time.

**Q**: "only measure the actual running time by performing 1000 steps gradient descent with the fixed length"

**A**: As described from Line 223 through 225, the *number of training steps* and *batch size* of the base setting are exactly the same as those in the original BERT model. Also, during pretraining, the input sequence always has a fixed length. Hence, the measured running times simply proportional to the total pretraining times for both models.

**Q**: "trading sequential resolution for more layer"

**A**: As we discussed in Section 2.3, reducing the sequence length will inevitably lead to capacity drop but the capacity drop can be compensated by re-investing the *saved FLOPs* in stacking more cheaper layers. Fundamentally, the key question is: Given the **same amount of FLOPs**, should we invest the FLOPs in (a) fewer full-length layers, or (b) more reduced-length layers? This work explores this fundamental question and shows that option (b) can empirically improve the performance. In comparison, simply increasing the layers of standard transformer will require much more FLOPs.

**Q**: "What is the significance of length reduction"

**A**: By comparing the performance between L12H768 and B4-4-4H768 and between L24H1024 and B8-8-8H1024, the current length reduction mechanism by itself can mildly harm the performance in certain tasks but achieve the same performance in others, with the benefit of over 40% fewer FLOPs. Hence, the ultimate effects are largely task-specific.

**Q**: "there should be more theoretical analysis and experiments on length reduction ..."

**A**: We agree that developing theoretical understanding of length reduction / compression is an important direction for future work. Meanwhile, we believe identifying the scientific possibility of the idea, designing a practically scalable instantiation, and empirically showing the potential of this direction are also valuable and necessary steps.

[Meta-Review · NeurIPS 2020]

This paper proposes a new model to extend Transformer. The key idea is to shrink the network to improve efficiency in computation. Strength • The proposed method is novel and technically sound. • Experiments have been conducted and the results are convincing. Weakness • The model is a little bit complex.